



# 1 Widespread increase in discharge from West Antarctic Peninsula
# 2 glaciers since 2018

Benjamin J. Davison[1], Anna Hogg[1], Carlos Moffat[2], Michael P. Meredith[3], Benjamin, J. Wallis[1]
[1]School of Earth and Environment, University of Leeds, United Kingdom
[2]School of Marine Science and Policy, University of Delaware, Newark, DE, USA
[3]British Antarctic Survey, Cambridge, United Kingdom
*Correspondence to*: Benjamin J. Davison (b.davison@leeds.ac.uk)
**Abstract.** Many glaciers on the Antarctic Peninsula have retreated and accelerated in recent decades. Here we show that there
was a widespread, quasi-synchronous and sustained increase in grounding line discharge from glaciers on the west coast of the
Antarctic Peninsula since 2018. Overall, west Antarctic Peninsula discharge trends increased by over a factor of three, from
0.5 Gt/y/decade during 2017 to 2020 up to 1.6 Gt/y/decade in the years following, leading to a grounding line discharge
increase of 7 Gt/y (7.4%) since 2017. The acceleration in discharge was concentrated at glaciers connected to deep, cross-shelf
troughs hosting warm ocean waters, and the acceleration occurred during a period of anomalously high subsurface water
temperatures on the continental shelf. Given that many of the affected glaciers have retreated over the past several decades in
response to ocean warming, thereby highlighting their sensitivity to ocean forcing, we argue that the recent period of
anomalously warm water was likely a key driver of the observed acceleration. However, the acceleration also occurred during
a time of anomalously high atmospheric temperatures and glacier surface runoff, which could have contributed to speed-up by
directly increasing basal water pressure and, by invigorating near-glacier circulation, increasing submarine melt rates. The
spatial pattern of glacier acceleration therefore provides an indication of glaciers that are exposed to warm ocean water at depth
and/or have active surface-to-bed hydrological connections. Both atmospheric and ocean temperatures in this region and its
surroundings are likely to increase further in the coming decades, suggesting that discharge increases may continue and become
more widespread.

## 23 1 Introduction

The Antarctic Peninsula (AP) hosts over 800 tidewater glaciers, which collectively hold an ice mass equivalent to 69±5 mm
of global sea level rise (Huss and Farinotti, 2014). Substantial changes in glacier and ice shelf area have occurred across the
AP since the mid-20th century (Cook and Vaughan, 2010; Doake and Vaughan, 1991; Rott et al., 1996). Many studies have
focused on changes to AP ice shelves, including the retreat of Wordie Ice Shelf from 1966 to 1989 (Doake and Vaughan, 1991;
Vaughan and Doake, 1996), Prince Gustav Ice Shelf during 1989 to 1995 (Cooper, 1997), Larsen-A in 1995 (Rott et al., 1996),
Larsen-B in 2002 (Rack and Rott, 2004; Scambos et al., 2003) and Wilkins Ice Shelf in 2008 (Braun et al., 2009). These
changes in ice shelf area have generally been attributed to rising surface air temperatures, leading to extensive melt ponding,





hydrofracture and rapid successive calving of elongate icebergs parallel to the ice shelf edge (Scambos et al., 2009). Glacier
acceleration and thinning has followed the collapse of these ice shelves due to loss of ice shelf buttressing – the Larsen-B
tributary glaciers have become a heavily researched example of this response (Rignot et al., 2004; Rott et al., 2018; Scambos
et al., 2004; Seehaus et al., 2018; Wuite et al., 2015). Although the well-documented initial acceleration and subsequent
deceleration of those glaciers was substantial, measurements of AP mass change over recent decades remain uncertain because
of very large uncertainties in bed elevation and surface mass balance (Gardner et al., 2018; Hansen et al., 2021; Rignot et al.,
2019; Rott et al., 2018), though recent efforts to downscale regional climate model output has led to significant improvements
(Noël et al., 2023).
Outside of ice shelf tributary glaciers, tidewater glaciers on the AP have received less research attention. The majority of such
glaciers on the west coast have retreated since at least the 1980s (Cook et al., 2005, 2014; Cook and Vaughan, 2010), seemingly
in response to increased flow of relatively warm (> 1°C) Circumpolar Deep Water (CDW) onto the continental shelf south of
Bransfield Strait (Cook et al., 2016). Glaciers in the southwest AP draining into the George VI Ice Shelf and Bellingshausen
Sea have accelerated (Hogg et al., 2017) and thinned (Wouters et al., 2015) since the late-2000s. In addition to these long-term
changes in area, speed and thickness, many glaciers along the west AP coast appear to undergo seasonal changes in ice velocity
(Boxall et al., 2022; Wallis et al., 2023b), which may be driven by changes in surface and upper-layer ocean temperature,
surface-derived meltwater flow at the ice-bed interface, changes in sea ice coverage or some combination thereof. Pulses of
meltwater supply to the ice-bed interface, caused by rapid supraglacial lake drainage or extreme melt events, may cause some
glaciers on the AP to undergo rapid, short-lived accelerations (Tuckett et al., 2019) but, insofar as they do occur, they remain
challenging to detect (Rott et al., 2020).
More recently, a large and sustained acceleration and retreat of Cadman Glacier on the west AP has been documented (Wallis
et al., 2023a). This acceleration and retreat began in 2018 during a period of anomalously high subsurface ocean temperatures
on the continental shelf, due to an incursion of warm CDW. Whilst the glaciers immediately adjacent to Cadman Glacier were
protected from this incursion of warm CDW by shallow sills, many glaciers on the west AP will not have such protective sills,
raising the possibility of a more widespread response of glaciers on the west AP. Identifying and attributing such a response
is important because understanding drivers of grounded ice speed change is informative for interpreting present-day glacier
mass changes and for reducing uncertainties in projections of future glacier mass change. In this study, we examine changes
in ice speed, grounding line discharge, terminus positions and ocean temperature along a substantial section of the west AP
(Figure 1) during this period of anomalously high atmospheric and subsurface ocean temperature.
**2 Methods**
**2.1 Grounding line discharge**
Grounding line discharge is the mass of ice crossing the point at which the glacier is last in contact with the underlying
topography as it flows seawards. In the case of tidewater glaciers with relatively stable termini, it approximates the calving



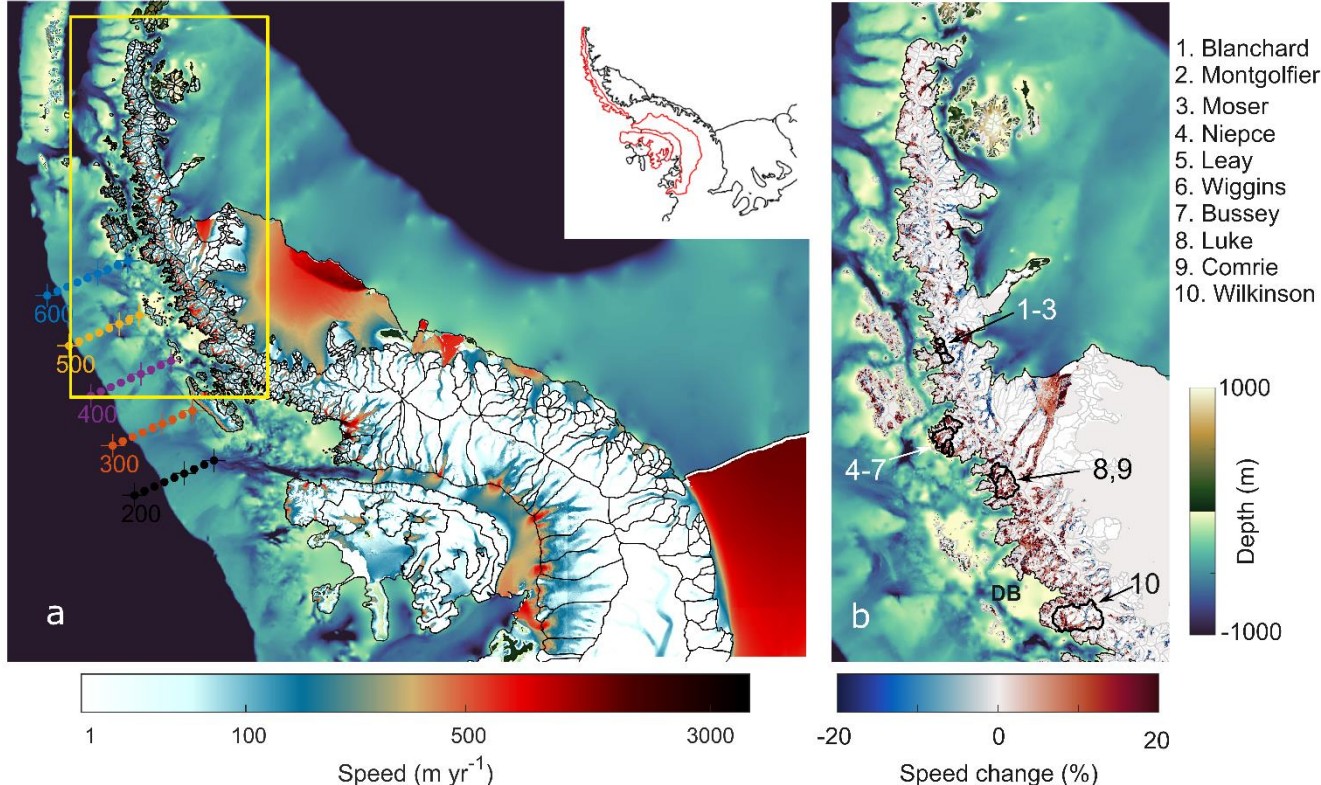

**Figure 1.** Study area overview. (a) April 2014 to April 2024 mean ice speed and bathymetry (Morlighem et al., 2020) of the Antarctic Peninsula. Routinely-repeated Conductivity-Temperature-Depth (CTD) stations from the Palmer Long-Term Ecological Research programme shown by numbered and coloured dots. The crossed dots indicate CTD stations acquired since 2009. Glacier drainage basins (Cook et al., 2014) are outlined in black and the inset shows basins Hp-I and West Graham Land outlined in red. (b) Ice speed change between the periods 2017/04/01-2020/09/01 and 2020/04/01-2023/09/01, as a percentage of the long-term average speed. DB in (b) indicates Darbel Bay.

flux. We use the monthly grounding line discharge dataset of Davison et al. (2023); readers are referred to that paper for full
methodological details. For the purposes of this study, we use the 'FrankenBed' version of the discharge dataset, which uses
a 100x100 m bedrock grid for the Antarctic Peninsula (Huss and Farinotti, 2014) and accounts for changes in surface elevation
over time using time-dependent polynomial fits to observed surface elevation changes posted on a 5x5 km grid at quarterly
intervals (Shepherd et al., 2019). During the study period (2017-2023), all the discharge estimates are calculated using 100x100
m velocity estimates derived from intensity tracking of Sentinel-1 6- and 12-day image pairs, making them particularly suitable
for resolving changes in speed on the relatively narrow outlet glaciers of the AP. The discharge dataset includes all glaciers
and basin definitions; in this study, we restrict our analysis to glaciers in the west AP, which we define as basins whose centre
coordinate falls within West Graham Land or basin Hp-I, as defined by Mouginot et al. (2017) (Figure 1).





## 2.2 Discharge change point

For each tidewater glacier basin on the west AP, we use change point analysis to identify the single most substantial change in grounding line discharge linear trends since 2017. Change points are defined as the time at which the discharge trends before and after the change point differ the most. To identify glaciers with a significant acceleration, we isolated basins where the discharge trend during the second period was positive, at least 50 % greater than during the first period and where the P-value of the trend during the second period was less than 0.1 – we chose not to restrict our analysis just to basins with more significant trends (e.g. P < 0.05) because of the short time periods over which trends were calculated. For all basins, we calculated the change in trend before and after the change point, in order to highlight glaciers that underwent a trend acceleration or even a trend reversal, from decelerating to accelerating. We excluded change points falling within 20 months (25 %) of the beginning or end of the study period, to minimise aliasing of seasonal discharge variability. In this study, 10 glaciers were selected for detailed examination, being the ones with the strongest changes in discharge trend and hence the ones from which the relevant dynamics are most likely to be ascertainable (Figure 1).

## 2.3 Terminus positions

For each of the 10 glaciers selected, we measure interannual changes in glacier terminus position by delineating termini in all available cloud-free Sentinel-2 imagery between February and May each year from 2016 to 2023. Higher frequency measurements show that there is seasonal terminus advance and retreat along the west AP, with the most advanced positions generally occurring at the end of the Austral winter and the most retreated positions occurring at the end of summer (Wallis et al., 2023b). By focusing on Sentinel-2 imagery from February to May, our measurements approximate the seasonally most retreated position whilst avoiding the difficulties posed by low radar backscatter during the melt events and by Digital Elevation Model artefacts that can affect Sentinel-1 Ground Range Detected imagery in this area of steep topography. We perform the terminus delineations in the Google Earth Engine Digitisation Tool (GEEDiT), and use the multi-centreline method in the Margin Change Quantification Tool (MaQiT) to calculate width-averaged terminus position change for each glacier (Lea, 2018). When calculating width-averaged terminus position change, we only include sections of the terminus delineated at every measurement epoch.

## 2.4 Atmospheric and ocean temperature change

We extract daily 2 m atmospheric temperatures over the west AP from 1979 through 2023 from ERA5 reanalyses (Hersbach et al., 2020) and calculate daily anomalies relative to the 1979-2008 daily climatology. We calculate ocean temperature anomalies along five Conductivity-Temperature-Depth (CTD) sections occupied during the Palmer Long-Term Ecological Research (LTER) programme (Smith et al., 1995). The Palmer LTER CTD dataset provides quasi-annual snapshots of conservative ocean temperature, typically during January, along transects from beyond the continental shelf break to near the west AP coastline. For this study, we selected the five transects occupied most frequently (locations in Figure 1), each separated



by approximately 100 km, extending from Marguerite Bay in the south to Palmer Basin in the north. In 2009, the Palmer-
LTER programme extended its sampling grid latitudinally but reduced its cross-shore resolution (Figure 1). Here, we calculate
conservative temperature anomalies during each cruise relative to the 1999-2008 mean for each transect, during which time
the programme was still using the high-resolution grid. We also examine daily runoff time-series from 5x5 km resolution
RACMO2.3p2 (van Wessem et al., 2018).

## 3 Results

### 3.1 Acceleration of grounding line discharge

We observe widespread changes in speed on the AP between the April 2017-September 2020 and April 2020-September 2023
periods (Figure 1b; Figure 2). The majority of tidewater glaciers draining the west AP accelerated by 5-20 % since April 2017,
leading to an overall 7 Gt (7.4 %) increase in west AP grounding line discharge. This was most pronounced in the fast-flowing
trunks of the larger outlet glaciers and was clearest at Montgolfier Glacier, Niepce Glacier, Luke Glacier, Comrie Glacier and
Wilkinson Murphy Glacier, where speeds increased by over 20 %  (Figure 2). At some glaciers, such as Blanchard Glacier and
Montgolfier Glacier, we observe slow-down in the shear margins and around high elevation ice falls (Figure 2b,c), which we
hypothesise is due to shear margin damage and dynamic thinning.
Throughout the observation period, grounding line discharge has increased at almost every glacier basin in the west AP. For
some basins, the discharge increase is relatively steady and is part of a longer-term trend. In this study, we focus on glaciers
that underwent a notable change in grounding line discharge trend between 2018 and 2021 (Figure 3). To illustrate, grounding
line discharge at Wilkinson Murphy Glacier remained steady at 2017 levels, with fluctuations of magnitude less than 5 % from
2017 to June 2020, after which discharge increased at a rate of 3.4 % yr$^{-1}$ to a maximum around 10 % greater than 2017 levels
(Figure 3j). Similarly, the positive trends in discharge at Montgolfier Glacier, Niepce Glacier and Luke Glacier all increased
by more than a factor of five between May 2021 and January 2022 (Figure 3b,d,h). Some glaciers, such as Moser Glacier,
Leay Glacier and Bussey Glacier transitioned from a period of weakly declining discharge to very strongly increasing discharge
during this broad period of acceleration (Figure 3c,e,g).
These large increases in linear discharge trends are widespread along the west AP (Figure 4). The majority of glaciers north
of Blanchard Glacier and south of Wilkinson Murphy Glacier generally had little change in discharge trend since 2017.
However, the majority of glaciers in the central west AP, between Blanchard and Wilkinson Murphy glaciers, exhibited a
significant increase in discharge trend, with significance determined as per above. There appears to be some clustering to the
discharge changes – some areas, such as Darbel Bay (location in Figure 1), host several glaciers that appear to have little
change in discharge. In the case of Darbel Bay, the bathymetry is shallow (<100 m based on BedMachine v3; Morlighem et
al., 2020), limiting the transport of warm CDW to the coast. However, other 'low responders' do not always coincide with
areas of shallow bathymetry and sometimes have responsive neighbouring glaciers. As in Wallis et al. (2023a), these cases



**Figure 2.** Speed change of selected glaciers between the periods 2017/04/01-2020/09/01 and 2020/04/01-2023/09/01. (a) Blanchard, (b) Montgolfier, (c) Moser, (d) Niepce, (e) Leay, (f) Wiggins, (g) Bussey, (h) Luke, (i) Comrie and, (j) Wilkinson Murphy. The background is a hillshade of the Reference Elevation Model of Antarctica 100 m mosaic (Howat et al., 2019).

may reflect the presence of shallow sills not captured by BedMachine v3, which would act as barriers to incursions of warm
water below the sill depth (Bao and Moffat, 2024).
There is broad consistency in the timing of discharge trend changes amongst west AP glaciers (Figures 4 & 5). A vast majority
of glaciers with significant discharge trend increases began to accelerate during the austral summer of 2020/2021 (Figure 5),
though there is spread around this period (Figures 3 & 4d). Prior to the change point for each glacier, there was a range of
discharge trends, with some glaciers decelerating, accelerating or remaining approximately steady with less discharge than in
2023 (Figure 5). Since the approximate time of that summer, however, there has been a widespread, quasi-synchronous
acceleration of glaciers along a large section of the west AP, leading to peak discharge at or near the end of our observations
in 2023 (Figure 5).





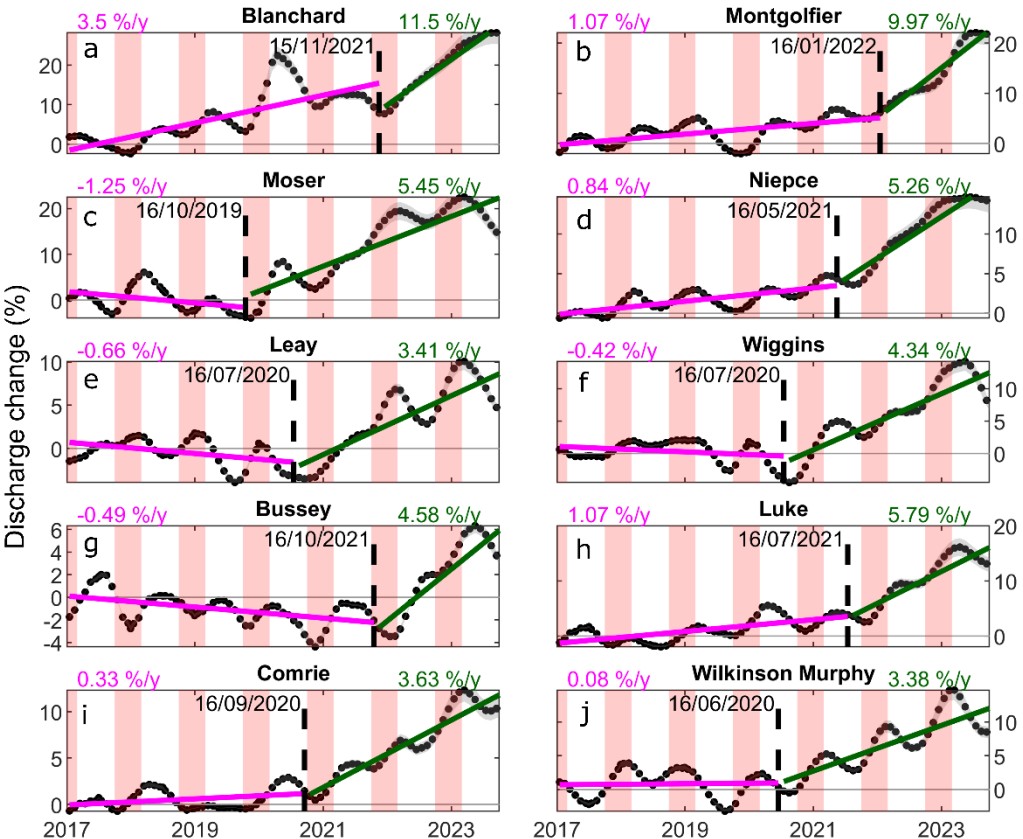

**Figure 3.** Grounding line discharge change at selected glaciers. In each panel, grounding line discharge change (relative to the 2017 mean) and associated error are shown as black dots and grey shading. The timing of the change in discharge trend is shown by the dashed line with the date labelled. The linear trends before and after the change point are shown in magenta and green respectively. The red shading indicates the Austral summer (December through February).

## 3.2 Terminus position change

We examined changes in terminus position at the end of the austral summer from 2016 to 2023 at our 10 example glaciers. Perhaps surprisingly, inter-annual terminus position changes at 7 of the 10 selected glaciers is negligible or not discernible from seasonal fluctuations in terminus position (not shown). Bussey Glacier exhibited modest but clear retreat of just 20 m on average and by 150 m on its true left margin (Figure 6). Wiggins Glacier experienced slightly greater retreat of over 100 m averaged across the width of the terminus and by approximately 240 m at the most affected section (Figure 6). Wilkinson Murphy Glacier retreated by 1 km on average since 2017 and by over 1.5 km across much of its fast-flowing centre (Figure 6). The timing of terminus position changes at these glaciers broadly coincide with the observed changes in grounding line discharge, with the majority of retreat occurring since 2019.



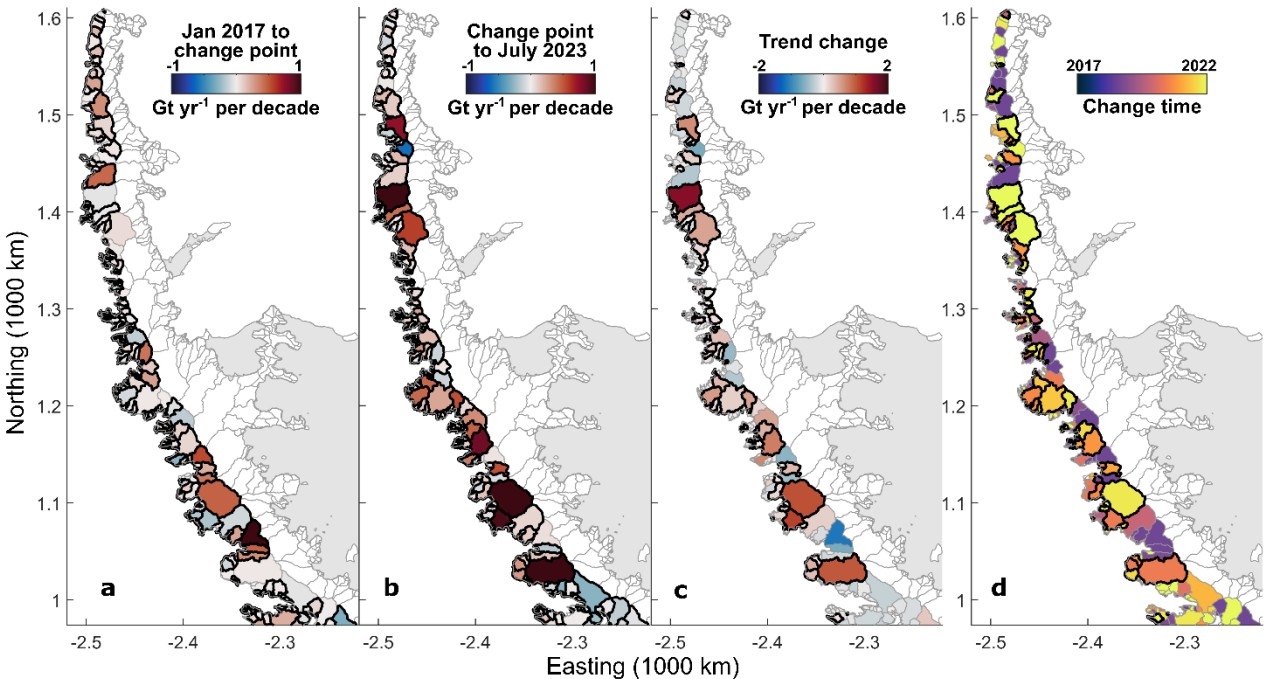

**Figure 4.** Overview of discharge trend changes. (a) Linear trend in discharge from January 2017 to the change point for each basin on the west coast of the Antarctic Peninsula. (b) Linear trend in discharge from the change point to July 2023. (c) The change in discharge trend before and after the change point, with positive values indicating a trend increase. (d) The timing of the discharge trend change. Basins with significant trends (P<0.05; a,b) or significant trend increases (see text for details; c,d) are outlined in black.

### 3.3 Ocean temperature change

The conservative temperature anomalies from the Palmer LTER CTD transects (locations in Figure 1) clearly show a warming trend on the west AP continental shelf below 100 m from 1993 to 2021, and a cooling trend above 100 m (Figures 7 & 8). The significant linear trends in water temperature across all transects range from 0.02 °C dec$^{-1}$ to 0.21 °C dec$^{-1}$. Of particular relevance to this study, from 2018 to 2021 there was a positive temperature anomaly at 100-200 m depth that built to a peak of over 1°C above the long-term average in December 2021, with an anomaly maximum around 100 m depth (Figures 7 & 8). There is variability superimposed on these trends; for example, there was a period of more rapid warming below 100 m during the 1990s. In addition, the summers of 2013 through 2017 were generally cooler than the summers of 2007 through 2009 along transect 200 (Figure 7). These patterns are well-documented by several other publications (e.g. Cook et al., 2016; Martinson et al., 2008) and the warm periods are associated with sea ice coverage changes and wind-driven CDW warming and shoaling within the Antarctic Circumpolar Current (Moffat and Meredith, 2018; Schmidtko et al., 2014), allowing more and warmer CDW to access the continental shelf.




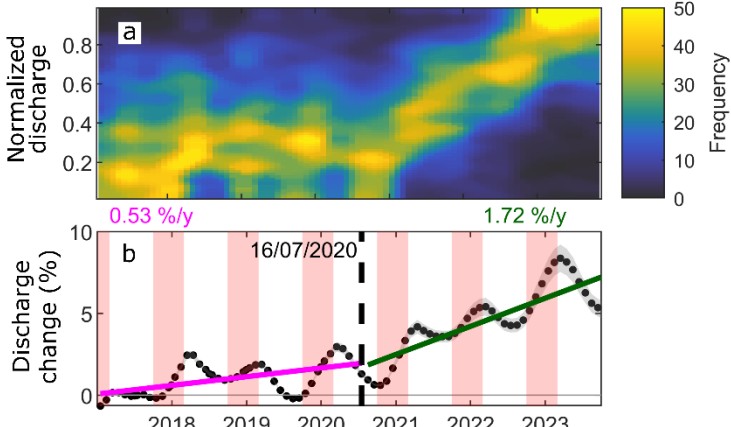

**Figure 5.** Discharge change across the west Antarctic Peninsula. (a) Frequency-density of normalized discharge time-series. Only west AP basins with a trend increase of more than 50% (N=97) were included to illustrate the synchronicity of the acceleration. (b) Grounding line discharge change (relative to the 2017 mean) of West Graham Land and associated error are shown as black dots and grey shading. The dashed line shows the timing of the change in discharge trend. The magenta and green lines show the linear trends before and after the change point. The red shading indicates the Austral summer (December through February).

## 4. Discussion


Many glaciers on the west AP have been retreating over recent decades (Cook et al., 2005). This retreat appears to have a
strong latitudinal pattern, with southern glaciers retreating faster, driven by a long-term increase in subsurface ocean
temperatures (Cook et al., 2016; Meredith and King, 2005), caused in turn by warming, shoaling and greater penetration of
CDW onto the continental shelf (Moffat and Meredith, 2018). In addition, many of the west AP glaciers are clearly responsive
to shorter-term changes in ocean temperature and, possibly, surface melt supply, resulting in seasonal changes in ice velocity
and terminus position (Boxall et al., 2022; Wallis et al., 2023b). Therefore, it is reasonable to assume that the west AP glaciers
could be responsive to multi-year anomalies in subsurface ocean temperature and/or meltwater supply. Our observations reveal
a widespread, quasi-synchronous and sustained increase in grounding line discharge across the west AP, centred around the
austral summer of 2021 (Figures 3-5). The response is concentrated in the central west AP, where warm CDW accesses the
glaciers via deep, cross-shelf troughs in the continental shelf. The majority of glaciers further north, which are not exposed to
CDW, exhibit muted or no change in grounding line discharge trends (Fig 4c). There is variability in the timing and magnitude
of glacier response along the coast, which will be governed by individual glacier geometry (Seehaus et al., 2018), proximal
fjord bathymetry (Bao and Moffat, 2024; Wallis et al., 2023a) as well as the competition between distinct processes (e.g. cross-
shelf transport and modification of CDW vs transport of cold water from the Weddell Sea around the tip of the Peninsula)
setting the subsurface ocean temperature (Moffat and Meredith, 2018; Venables et al., 2017). In places, this results in very
different responses between neighbouring glaciers and, for some glaciers, a continuation of their longer-term discharge trends
(Figure 4).



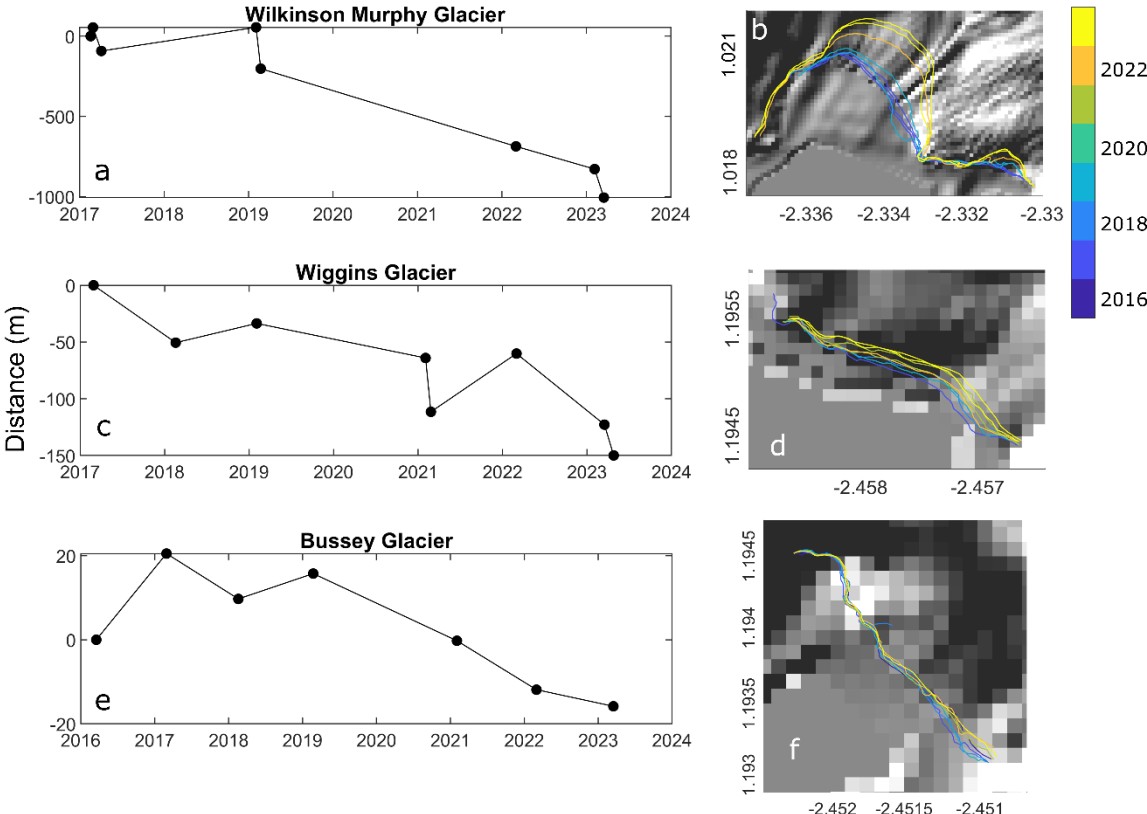

**Figure 6.** Overview of terminus position changes at four of the selected glaciers. The left column (a, c, e) show width-averaged terminus position change relative to the first measurement. The right column (b, d, f) illustrates the location of the terminus at each measurement time, overlaid on a hillshade of the Reference Elevation Model of Antarctica 100 m mosaic (Howat et al., 2019). The units in (b), (d) and (f) are 1000 km and the projection is South Polar Stereographic (EPSG 3031).

The widespread, quasi-synchronous and sustained nature of the discharge change points to a regional, sustained forcing. The
hydrographic observations show that there was a widespread and coherent increase in subsurface ocean temperatures on the
continental shelf from 2018 onwards, centred at 100-200 m depth and extending to the ocean bed on the continental shelf
(Figures 7 & 8). We do not have observations from the waters immediately adjacent to any of the west AP tidewater glaciers,
so we do not have direct evidence that the anomalously warm waters came into contact with the tidewater glaciers and elevated
submarine melt rates. However, the Palmer LTER data indicate that anomalously warm modified CDW was present across the
continental shelf south of Bransfield Strait during the 2018-2021 period, including in the deep, glacially-carved troughs that
connect the shelf edge to the west AP glaciers (Arndt et al., 2013; Cook et al., 2016; Couto et al., 2017). In addition, diverse
local CTD measurements along the west AP have documented the presence of CDW in immediate proximity to glacier termini
in the same region (Meredith et al., 2022; Venables et al., 2023), demonstrating that CDW does penetrate to parts of the coast.
It is therefore highly likely that the anomalously warm water present on the continental shelf from 2018 to at least 2021 came
into widespread contact with the west AP glaciers south of Bransfield Strait.





**Figure 7.** Conservative temperature anomalies relative to the 1999-2008 mean along transect 200. The vertical grey dashed lines indicate individual cast locations – note that the panel outlines obscure casts at the transect endpoints. The dark grey shading is topography from BedMachine v2 (Morlighem et al., 2020) and the Antarctic Peninsula coast is on the right.

Assuming that this contact did happen and that there was no commensurate drop in current velocity at the ice-ocean interface,
we would expect terminus submarine melt rates to increase. Glacier terminus depths along the west AP are poorly mapped,
but the available data indicate that many glaciers are several hundred metres thick at the terminus (Arndt et al., 2013; Cook et
al., 2016). Glaciers with grounding lines deeper than 100 m would be exposed to the anomalously warm CDW during each
Austral summer since 2018, likely leading to enhanced undercutting. The temperature anomalies were greatest around 100-
200 m depth; therefore, the enhancement of undercutting would lead to more pronounced quasi-linear or step-like undercuts





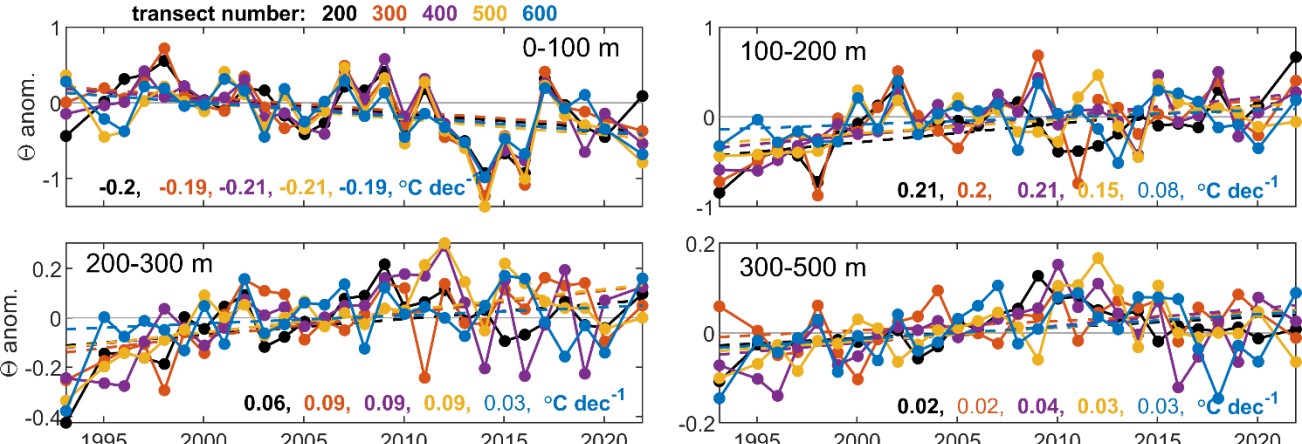

**Figure 8.** Conservative temperature anomaly time-series. Each panel illustrates time-series of conservative temperature anomalies within the given depths for each transect. The robust linear trends in temperature for each transect are quoted and significant trends (P<0.05) are in bold.

for glaciers shallower than 200 m and parabolic undercuts for more deeply grounded glaciers. Comparable undercut profiles
have been observed at glaciers in Greenland in the presence of similar vertical temperature profiles (Fried et al., 2015; Rignot
et al., 2015).
The majority of theoretical and numerical perspectives (Benn et al., 2017; Krug et al., 2015; Ma and Bassis, 2019; O'Leary
and Christoffersen, 2013; Slater et al., 2021) suggest that such profiles of undercutting can amplify calving, leading to retreat
and glacier acceleration. We observe retreat at just three of our ten example glaciers, only one of which (Wilkinson Murphy
Glacier) was very substantial. We do not have terminus position measurements at the tens of other west AP glaciers that
accelerated since the Austral summer of 2020/2021. In the absence of terminus retreat, more rapid submarine melting must be
balanced by faster ice velocities (Krug et al., 2015), such that the position of the calving front becomes a function of the
velocity and thickness of the upstream ice, rather than the driver of upstream ice velocity changes (Benn et al., 2007).
If enhanced submarine melting were the primary driver of the glacier acceleration, then the spatial pattern of glacier
acceleration provides information about the pathways by which the warm water accessed the west AP coastline. Most of the
glaciers that accelerated were located between Adelaide Island and Anvers Island, where several deep troughs provide a direct
pathway across the shelf along which CDW intrusions can access the central west AP (Arndt et al., 2013; Cook et al., 2016;
Couto et al., 2017). Some glaciers, such as Blanchard Glacier, located further north, where CDW influence on deep water
temperatures is at least seasonal (Wang et al., 2022), also accelerated. Such instances likely reflect the convoluted topographic
routes that dissect the west AP shelf and the competition between CDW and Weddell Sea waters on deep water temperatures,
among other processes. The majority of the northern-most glaciers along the WAP, which drain into Bransfield Strait and are
not exposed to warm CDW, showed weak or no acceleration. In addition, we observe acceleration at some glaciers that,
according to bathymetry products (Morlighem et al., 2020), are grounded in shallow water. For example, Luke Glacier and



Comrie Glacier (locations in Figure 1) are essentially land-terminating in BedMachine v3 yet are several hundred metres thick
in an independent thickness product (Huss and Farinotti, 2014). These and other similar sites may therefore indicate regions
to target in future bathymetric mapping efforts, or at least for improvement in future bed topographic assimilation efforts.
At most depths along the central west AP continental shelf, the conservative temperature anomalies since 2018 were similar
to or slightly larger than during warm periods in the late-2000s (Figure 8), so it is possible that ocean forcing alone was not
sufficient to drive the observed acceleration. In addition to warming ocean waters, ERA5 atmospheric temperatures over the
west AP have been anomalously high persistently since 2016 (Figure 9a). There were record high atmospheric temperatures
over the AP in February 2020 and 2022 (Francelino et al., 2021; Gorodetskaya et al., 2023). These heatwaves caused record-
high levels of snowmelt and rainfall (Gorodetskaya et al., 2023) that in turn led to extreme melt ponding, for example on the
George VI and Larsen-C ice shelves in 2020 (Banwell et al., 2021; Bevan et al., 2020). Output from RACMO2.3p2 (van
Wessem et al., 2018) - a 5.5 km regional climate model - shows that there is a modest amount of runoff (i.e. snowmelt that
does not refreeze in the firn) from the west AP (Figure 9b. The presence of plumes along the west AP coastline (Rodrigo et
al., 2016) provide strong evidence that at least some of this surface-derived meltwater and runoff does reach the ice-bed
interface and is discharged at the grounding line. Theoretical perspectives (e.g. Jenkins, 2011; Slater et al., 2016) and numerous
observational and modelling studies from other regions (e.g. Jackson et al., 2017; Sutherland et al., 2019; Straneo et al., 2011;
Carroll et al., 2016) show that the turbulent mixing and entrainment induced by subglacial discharge-driven plumes increases
glacier submarine melt rates. The RACMO2.3p2 runoff data indicate that runoff was much higher during February 2020 and
2021 than during the preceding years; this would drive more vigorous plumes and faster submarine melt rates, potentially
amplifying the effect of the observed warmer subsurface waters (Slater and Straneo, 2022).
In addition to modifying submarine melt rates, surface-to-bed meltwater injection could directly increase glacier speeds
through two mechanisms. If ice at some areas of the bed is below the pressure melting point, as some models indicate for the
AP (Dawson et al., 2022), and the surface-derived meltwater refreezes at the bed, the resulting release of latent and sensible
heat would raise the temperature of the ice – a process called cryohydrologic warming - thus causing the ice to deform more
rapidly. This process has been inferred at high-elevation areas of the Greenland Ice Sheet and linked to persistent acceleration
(Doyle et al., 2014). In addition, surface-to-bed meltwater injection to the bed can raise basal water pressure and transiently
increase basal sliding rates. There is some evidence of this processes on the AP over weekly to seasonal time-scales (Boxall
et al., 2022; Tuckett et al., 2019; Wallis et al., 2023b). There is exhaustive evidence that such meltwater-induced accelerations
on other ice masses have little impact on annual ice displacement, because of meltwater-induced compensatory periods of
slower ice flow (e.g. Sole et al., 2013). That may also be the case on the AP; however, there are no direct observations of
meltwater-induced changes in ice velocity on the AP to demonstrate that the same compensatory subglacial drainage
mechanisms operate here. It is possible that the combination of moderately thick, fast-flowing ice, low meltwater supply and
potentially extensive firn aquifers (Van Wessem et al., 2021) may result in qualitatively different meltwater-induced ice
velocity changes compared to those observed elsewhere. In addition, the extreme meltwater production in 2020 and 2022 may



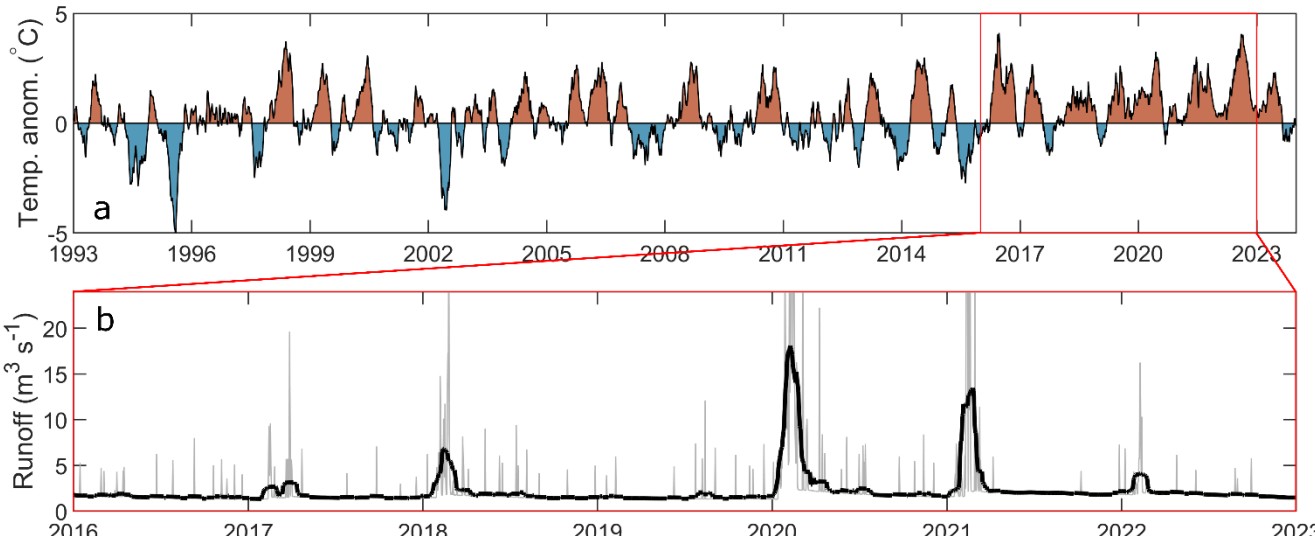

**Figure 9.** Atmospheric conditions over the west Antarctic Peninsula. (a) 2 m atmospheric temperature anomalies relative to the 1979-2008 daily climatology over the west AP from ERA5 reanalysis. The anomalies are smoothed with a 90-moving window. (b) Modelled runoff from a 5x5 km run of RACMO2.3p2, integrated over the west AP, from 2016 to 2023. Daily runoff is plotted grey and 30-day smoothed runoff in black. Panel (a) was plotted using the anomaly function in MATLAB (Greene, 2024).

have reduced firn pore space, allowing more surface-derived meltwater to penetrate to the ice-bed interface in subsequent,
lower melt years.
The widespread increase in grounding line discharge of the west AP observed here has implications for glacier mass balance.
Although the glaciers on the AP are small compared to their neighbours in other parts of West Antarctica, they are changing
rapidly such that AP contributed 14 % of Antarctica's total mass loss from 1992 to 2020 (Otosaka et al., 2023). Previous work
has linked warming subsurface ocean waters to widespread glacier retreat along the west AP (Cook et al., 2016) and more
recent work has further shown an ocean-driven ice tongue collapse and acceleration of Cadman Glacier on the west AP (Wallis
et al., 2023a). The observations presented in this study develop this understanding by showing a widespread, quasi-
synchronous acceleration of grounding line discharge along the west AP linked to a period of high air and subsurface ocean
temperatures. Unless surface mass balance increased commensurately, this recent acceleration of west AP glaciers will
accelerate the rate of west AP mass loss, contributing to faster rates of sea level rise. In addition, the increase in grounding line
discharge constitutes an increased solid freshwater input to the Bellingshausen Sea, which numerical modelling suggests can
increase ocean heat transport to West Antarctic ice shelves, leading to faster submarine melt rates (Flexas et al., 2022).



## 5. Conclusions

During the past half-century, tidewater glaciers on the west coast of the Antarctic Peninsula have retreated in response to rising subsurface ocean temperatures and they remain responsive to seasonal changes in atmospheric and ocean temperatures. This study identifies a widespread, quasi-synchronous and sustained increase in grounding line discharge of many glaciers along the west coast of the Antarctic Peninsula around the 2020/2021 austral summer. In many cases, grounding line discharge trends more than doubled and led to 5-20 % increases in grounding line discharge over a 2.5 year period. The acceleration of grounding line discharge occurred at a time of anomalously high, though not exceptional, subsurface ocean temperatures on the continental shelf, which would have increased terminus submarine melt rates and could have driven the observed glacier acceleration. The co-occurrence of record-high air temperatures and surface melting may have contributed to the glacier acceleration by increasing surface-to-bed meltwater delivery, potentially amplifying submarine melt rates and directly increasing glacier sliding speeds. In the absence of *in-situ* observations on the glacier surface and in the waters immediately adjacent to glacier calving fronts, there remain many uncertainties regarding the chain of events leading to this period of glacier acceleration, but we are hopeful that future campaigns to improve seafloor mapping, acquire near-glacier hydrographic measurements and to measure glacier velocity *in-situ* will provide important new understanding of the processes driving changes in ice flow on the Antarctic Peninsula. Nevertheless, it is clear that the recent period of anomalous atmospheric and ocean temperatures have, together or in isolation, driven a widespread and sustained acceleration of many west AP glaciers. We therefore speculate that, as the atmosphere and ocean in the region continue to warm, we are likely to see further coherent increases in grounding line discharge along the west AP with worsening implications for glacier mass balance, sea level rise and ocean circulation.

*Data availability.* The grounding line discharge dataset are available on Zenodo (https://zenodo.org/records/10417864). The Palmer LTER dataset were compiled for a previous study and made available on Zenodo (https://zenodo.org/records/10009821).

*Author contributions.* BJD conceived the study, performed the analysis and wrote the manuscript. BJW and CM compiled the raw CTD data into a format more amenable for analysis. All authors discussed the results and implications, and contributed to the manuscript preparation.

*Competing interests.* The contact author has declared that none of the authors have any competing interests.

*Acknowledgements.* We are grateful to the creators of the open-access satellite imagery, datasets and tools used in this study, and to the crews of all cruises involved in producing the Palmer LTER dataset. Data processing was undertaken on ARC3 and ARC4, part of the high-performance computing facilities at the University of Leeds, UK.



299

*Financial support.* BJD and AEH are supported by ESA through the Polar+ Ice Shelves project (ESA-IPL-POE-EF-cb-LE-2019-834) and the SO-ICE project (ESA AO/1-10461/20/I-NB), by NERC via the DeCAdeS project (NE/T012757/1) and by the UK EO Climate Information Service (NE/X019071/1). BJW is supported by the Panorama NERC Doctoral Training Partnership under grant NE/S007458/1. The Palmer LTER program is supported by NSF-OPP Grant #2026045.

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
