# Peer review of "Widespread increase in discharge from West Antarctic Peninsula glaciers since 2018"

_EGUsphere, 2024_

## Author Comment (AC1)

**Overview of changes**

We would like to thank the reviewers for providing thorough and constructive reviews of our manuscript. We overwhelmingly agreed with their comments and have revised our manuscript accordingly.

In summary, our major revisions include:

(1) Revision of our conclusions regarding the potential future behaviour of the west AP glaciers
(2) More robust change point analysis that is less sensitive to seasonal variations in discharge, and improved clarity throughout the manuscript regarding which glaciers were included in the analysis
(3) A new boxplot figure to complement the information shown in map-form in Figure 4

In addition, we have made a number of minor or technical corrections.

Below, we have reproduced the comments from both reviewers in black, with our response in blue. We have also appended a copy of the manuscript with the proposed changes marked.

**Responses to comments from Reviewer #1**

Summary: The authors apply change point analysis to an existing ice discharge dataset (created by the lead author) and then compare the results to time series of terminus position and environmental change. ERA5 air temperatures and CTD time series from the Palmer LTER are used to construct records of environmental change. The authors find that discharge increased around 2018, coinciding with both anomalously warm ocean and air temperatures. However, there is variability in the timing and magnitude of discharge change, such that it is difficult to determine the exact cause of the discharge variability. Still, the authors suggest that the widespread increase in discharge that is observed in recent years is an indicator of ongoing and future sensitivity to climate change across the region.

The paper is well written and fairly easy to read, with the exception of a few minor points described below. I appreciate the use of example glaciers but particularly like the map figures for all glaciers in the region (Figure 4) since it clearly shows variability in discharge change that is difficult to decipher from the example glaciers. Most recommended revisions are minor in nature, with the exception of a few points regarding the way that the change in discharge is described and the discussion of the change with respect to environmental forcings.

We are grateful to the reviewer for their thorough and constructive review of our manuscript. We overwhelmingly agree with their comments, and we have revised our manuscript accordingly. In particular, we have (1) included new figures to illustrate the change in discharge trend along the west coast of the AP (box plots/histograms); (2) included a more robust analysis of trend change that quantifies the impact of aliasing seasonal discharge changes; this analysis affects the number of glaciers detected with significant trend changes, but does not affect our conclusions; (3) revised our wording regarding potential future changes in discharge, given the uncertainty in the forcing and the apparent affect of glacier specific factors that will evolve as individual glaciers retreat or advance in future; (4) included more discussion of the spatial and temporal variability in discharge change shown in Figure 4.

Major Comments:

1.  I appreciate the use of change point analysis to identify changes in linear trends in discharge in the dataset because it minimizes user bias in the trend interpretation. That said, I think that the way the trends are discussed is a bit confusing/misleading at times. The time series seems quite short – 2017-2023 – yet the authors interpret the trends over as little as ~2 years to be indicative of longer-term changes. This might not be the authors' intention but presentation of trends over time as Gt/y/decade implicitly implies that the trends will persist for decadal time scales. Why not present the trends as Gt/y^2 as has been done elsewhere? Additionally, the authors point to the observed variability in discharge trends as an indicator of differences in sensitivity to environmental forcings that is at least partially due to differences in glacier geometry but then conclude that "discharge increases may continue and become more widespread". The observed variations in discharge change for individual glaciers and the apparent dependence on geometry means that the observed years-long changes in discharge will likely not persist on decadal time scales at individual glaciers or across the entire region in the coming decades. I recommend carefully revising wording to not make to many large leaps in interpretation of discharge change over the coming decades at the individual and regional scales.

    Good point regarding the trend unit. We included trends in Gt/yr/decade in the original manuscript because the trends in Gt/yr/yr are small (though still statistically significant), so presenting them in Gt/yr/decade made them more readable. We now present them as Mt/yr/yr, which is both readable and avoids implying longer term evolution.

    We agree with the reviewer's comment regarding the inconsistency and overzealousness of our interpretation regarding the future behaviour of these glaciers. We have revised our wording throughout the manuscript accordingly, instead focusing on the poorly understood spatial variability

and temporal variability in glacier response, and therefore the need for further investigation, given that atmospheric and ocean warming are likely to continue.

2. I think the use of example glaciers is really helpful when including such a large sample size in an analysis. That said, upon careful reading of the methods, I started to wonder if the ten glaciers that you focus on are used for more than just demonstration. On lines 81-82 you say they were selected for detailed examination because they have the strong changes in discharge trends. Does that mean the interpretation of "regional" change throughout the rest of the manuscript is entirely based on analysis of those 10 glaciers? Or are those glaciers simply used to emphasize the potential for large change? If you only analyzed data for those ten glaciers, then that needs to be made much more clear throughout the paper because you are not really performing a regionally-representative analysis if you are focusing on glaciers with end-member change.

We apologise for not making it clearer in the manuscript which glaciers were used to support our interpretation. Throughout the manuscript, all statistics draw on all glaciers shown in Figure 4, and we have revised the wording in the manuscript to make this clearer. The 10 example glaciers shown in Figure 2 are only used to illustrate what the trend change 'looks like' at the glaciers with the largest trend changes.

3. Cryohydrologic warming seems like an extremely unlikely cause for the observed changes in discharge. It is true that refreezing meltwater can increase deformation rates but I find it unlikely that it could cause 100s of meters of added deformation each year (inferred from Figure 2). I recommend that you either add estimates of cryohydrologic warming-enhanced flow for other locations from the literature and a comparison with changes in speed along the western AP, or you remove the mechanism as an explanation for enhanced discharge.

We have removed cryohydrologic warming as a suggested mechanism contributing to the observed increase in glacier speed.

Minor Comments:

- I don't think you need to describe the discharge dataset in great detail because it is already described in a published paper, but a brief description should be included to make it easier for the reader to interpret this paper. This statement is particularly true given that the referenced paper is still in discussion. Some basic details like the locations of the gates, the number of study sites, and any bias estimates or corrections are needed here. For example, what does "all glaciers and basin definitions" on line 70 mean?

  We agree our description in the initial manuscript was too brief. We have revised this section so that it now includes a description of the number and location of flux gates, some more detail on the applied corrections and more description of the glacier basins used in this study.

- Is there a limit to the number of observations over which the trends in discharge can be calculated? Or did you only limit the time period over which you will accept change points (20 months cut from each end)? How/why did you decide to exclude 20 months in particular? Did you force the trendlines to include full years of data to prevent the trends from becoming amplified if fit to partial years (for example an austral summer minimum, across a full year, to an austral winter maximum)?

  The trends in discharge can be calculated over a maximum of 88 observations (leaving 20 or more observations at either end). We decided to set 20 months/observations as the minimum to reduce aliasing of seasonal signals (our study period ends after the austral summer and typical timing of peak summer velocity). However, to further reduce seasonal aliasing, we performed a sensitivity analysis by re-calculating the trend change for all glaciers after shifting the change point by +/- 3 months in increments of 1 month. We only retained glaciers that exhibited a >=50 % increase in

discharge trend for all 7 possible change points, resulting in 42 glaciers identified as having a significant and sustained increase in grounding line discharge. We now also include some statistics and one figure to compare this set of glaciers with all other glaciers on the west AP. We have modified our wording throughout the manuscript to clarify which population of glaciers we are referring to.

- I really like Figure 4 but I'd also love to see histograms or box plots of the trends before and after the change point as well as the timing of the change point. Those figures would make it easier to determine synchronicity/uniformity in the data.

  This is a really helpful suggestion. We have included a new figure (Figure 5) in which the data in Figure 4 are presented as box plots. We think this illustrates how the 42 glaciers with significant change are distinct from most of the glaciers on the west AP and shows that the majority of those 42 glaciers accelerated around 2021.

**Responses to comments from Reviewer #2**

General comments

Davison et al. combined several datasets for glacier discharge, air temperature from ERA5, modeled runoff from RACMO, and manually-delineated terminus positions to identify and analyze a quasi-synchronous increase in discharge for glaciers on the West Antarctic Peninsula since 2018. They used change point analysis to identify the timing and relative magnitude of changes in discharge throughout the region.

Overall, the paper is well written and easy to follow. The authors provide a thoughtful discussion of mechanisms and uncertainties for attributing the regional speed up to ocean and air temperatures, and target this area for future bathymetric / bed mapping for better understanding, particularly related to submarine melt processes. I think this is a valuable contribution to the broader Antarctic and glaciological scientific communities. I have a few technical suggestions for improving the clarity of the text, noted below.

We thank the reviewer for their constructive feedback on the manuscript. We overwhelmingly agree with each of the comments and have implemented all of them (bar the last technical correction) in the revised version of the manuscript.

Technical corrections

- L18: "near-glacier circulation" - add "ocean" or "water" Done

- L61: "Ground line discharge is the mass…" Nitpicky note here: I suggest adding something to indicate the rate or flux of mass. It sounds more like a static variable in this sentence. Or simplify to e.g., "...the rate of mass flowing across the glacier grounding line towards the sea." Yes, we tied ourselves in knots a bit there. We have rephrased as suggested.

- Figure 1. Nicely laid out figure. In panel (a), I suggest using a sequential, more colorblind friendly colormap for the Speed variable, particularly to better distinguish it from the Speed change colors in panel (b). We have changed this to the 'thermal' colormap in the cmocean collection of colormaps, which are all perceptually uniform.

- L74: "Change points are…" I find this sentence confusing, can you rephrase? I think by "discharge trend" you mean the linear trend or regression coefficient, but it would help to be more specific. We have rephrased this section in response to this comment and a similar comment from another reviewer.

- Figure 4: Really nice figure! This is very helpful for interpreting your results. Thanks!

- Figure 6: Consider a different basemap e.g., LIMA or Sentinel-2 images for the map view panels. REMA is very blurry esp. for Wiggins and Bussey glaciers and not particularly useful for interpretation. We have revised this figure and now use the 15x15 m LIMA product as a basemap.

- L210: "was" We think "were" is correct

[revised manuscript text omitted]

---

## Author Response (AR1)

Dear Dr McCormack,

Thank you for checking our revised manuscript and for your requests. We have modified the new Figure 5 so that it presents the data in violin plots – we think this is a great improvement, so thanks especially for this suggestion. We have checked our figures in the Coblis colour blindness simulator as requested and found that the interpretation of the figures is not sensitive to colour blindness. The individual lines in Figure 9 are a little hard to differentiate with some types of colour blindness; however, the overall trends amongst all transects is key in this plot, which is still not affected by colour blindness, so we hope this is acceptable. We have expanded our data availability statement to include all datasets used in the manuscript.

We hope that you find our revised manuscript suitable for publication, but please let us know if you need us to make any further changes.

All the best on behalf of the authors,

Benjamin Davison

University of Sheffield